



# *Environment90m* - globally standardized environmental variables for spatial freshwater biodiversity science at high spatial resolution

Jaime R. García Márquez[1], Afroditi Grigoropoulou[1], Thomas Tomiczek[1], Marlene Schürz[1,2], Vanessa Bremerich[1], Yusdiel Torres-Cambas[1], Merret Buurman[1], Kristi Bego[1], Giuseppe Amatulli[3,4], and Sami Domisch[1]

[1]Leibniz Institute of Freshwater Ecology and Inland Fisheries, Department of Community and Ecosystem Ecology, Müggelseedamm 310, 12587 Berlin, Germany
[2]Freie Universität Berlin, Department of Biology, Chemistry, Pharmacy, Institute of Biology, Königin-Luise-Str. 1-3, Berlin, 14195 Germany
[3]Yale University, School of the Environment, 195 Prospect Street, New Haven, CT, 06511, USA
[4]Spatial Ecology, 35A, Hazlemere Road, Penn, Buckinghamshire, HP10 8AD, United Kingdom.

**Correspondence:** Jaime García Márquez (jaime.marquez@igb-berlin.de), Sami Domisch (sami.domisch@igb-berlin.de)

**Abstract.** The current loss of freshwater habitats and biodiversity calls for an immediate mobilization and application of existing data and tools to contribute to the development of sound strategies for their long-term conservation. However, one particular challenge for obtaining a baseline regarding the spatial distribution of freshwater habitats and biodiversity is the need for standardized high-resolution environmental information, which ideally can provide a characterization of freshwater habitats anywhere in the world. To address this challenge, we present the *Environment90m* dataset which aggregates a large number of environmental layers into each of the 726 million sub-catchments of the *Hydrography90m* dataset, corresponding to single stream segments. Specifically, *Environment90m* includes 45 variables related to topography and hydrography, 19 climate variables for the observation period of 1981-2010, as well as projections for 2041-2070 and 2071-2100 under the Shared Socioeconomic Pathways (SSPs) 1.26, 3.70 and 5.85, and three global circulation models (UKESM, MPI and IPSL). Moreover, *Environment90m* includes 22 land cover categories for the annual time-series data from 1992-2020. In addition, we provide 15 soil variables and information on aridity and modelled streamflow. Summary statistics (i.e., mean, min, max, range, sd) are provided for all continuous variables while for categorical data, the proportion of each category is calculated within each of the sub-catchments. The data is available at https://hydrography.org/environment90m. To facilitate data download and processing, we provide dedicated functions within the hydrographr R-package. For all underlying calculations, we used the open-source tools GDAL/OGR, GRASS-GIS and AWK, so that custom data can be easily generated using the hydrographr R-package. *Environment90m*, along with the tools, provides an array of opportunities for research and application in spatial freshwater biodiversity science, specifically biogeographical analyses and conservation in freshwater ecosystems.

## 1 Introduction

Freshwater biodiversity is among the terrestrial and marine realms most at risk WWF (2020); Tickner et al. (2020). Advances towards the protection of freshwater biodiversity, and consequently also freshwater habitats in general, remain elusive despite





the recent efforts towards the so-called "30-by-30" protection target which aims to protect 30% of Earth's lands, oceans, coastal areas and inland waters (The Post-2020 Global Biodiversity Framework, Hughes (2023)), or, the recent EU Nature Restoration Law that aims to restore river connectivity (Stoffers et al. (2024)). With high-level political intentions in place, large-scale and standardized analyses regarding the spatial distribution of freshwater habitats, their environmental characterization and connectivity, as well as biodiversity assessments are required to allow answering the question, which areas should be prioritized for protection? Addressing this goal requires at minimum a baseline regarding detailed knowledge of the spatial distribution of the environmental characteristics of freshwater habitats. This is because knowledge on the spatial distribution of specific freshwater habitats and their environment allows in turn to perform freshwater biodiversity assessments using e.g. distribution modelling techniques (Bellin et al., 2022), and to perform connectivity-related analyses to restore free-flowing rivers (Hermoso, 2025).

In the freshwater realm, information on such environmental characteristics should ideally be available at very high spatial resolution so that (i) it can be attributed to the corresponding water body, e.g., a specific segment of the river network, such that (ii) environmental characteristics are not aggregated across large areas given the Modifiable Area Unit Problem (MAUP) Jelinski and Wu (1996). The MAUP is a statistical feature which occurs when data is aggregated to spatial units, where the size of the units may influence the aggregation values (e.g. by using grid cells of varying size). In the freshwater realm, spatial units often correspond to drainages, larger sub-catchments or standing water bodies such as lakes, and environmental information is commonly aggregated across these units. A key goal is therefore to have environmental information at the highest possible spatial resolution while still achieving computational efficiency, allowing to incorporate the longitudinal connectivity which consists of a unique feature in the freshwater realm. The environmental conditions along the dendritic network structure can be depicted following the River Continuum Concept Vannote et al. (1980), macrosystem theory Thorp (2014) or functional process zones Maasri et al. (2019), and similarly, tributary inputs, lateral connectivity with floodplains, and discontinuities caused by natural or anthropogenic disturbances also play a role in shaping the environmental conditions along the dendritic stream network Stanford and Ward (1983); Ward and Stanford (1995); Benda et al. (2004). This requires to pinpoint the relevant environmental conditions and processes to single network segments. Following the MAUP, aggregating environmental characteristics across large drainage basins or catchments would lump the data, and challenges the attribution of specific environmental characteristics to these segments or water bodies Hermoso and Kennard (2012). Hence, the spatial aggregation of environmental information, which usually comes in gridded datasets at e.g. 1 km spatial resolution, has to match the spatial configuration of the water bodies Brunner et al. (2024); Friedrichs-Manthey et al. (2020). In this regard, sub-catchments which correspond to the single stream segments are, unlike pixels, non-randomly distributed across the surface and follow the topographical and topological gradients in the landscape Brunner et al. (2024). Sub-catchments consist therefore of the natural units in freshwater ecosystems and allow encompassing also riparian areas and aquatic-terrestrial linkages Linke et al. (2007). They feature the same connectivity as the network, but also allow including the terrestrial landscape into the analysis workflow, which is of interest when performing biogeographic analyses of e.g. aquatic insects, amphibians or mammals relying both on the aquatic and terrestrial realms.





Aggregating environmental variables across sub-catchments is often done on case-by-case basis, and only few datasets exist that cover wide spatial gradients. A global extent is given by the HydroATLAS Linke et al. (2019) and the HydroLAKES datasets Messager et al. (2016); **?**, as well as a near-global aggregation of upstream-catchment variables Domisch et al. (2015) which follow the HydroSHEDS river network Lehner et al. (2008). National efforts are provided by the StreamCAT Dataset Hill et al. (2016) and LakeCAT Dataset Hill et al. (2018) which correspond to the high-resolution NHDPlusV2 river network

McKay et al. (2012) in the United States.

To facilitate globally standardized analyses and modelling workflows, we introduce the new *Environment90m* dataset that aggregates environmental variables at global scale and at very high spatial resolution based on the Hydrography90m stream network Amatulli et al. (2022). Hydrography90m is a global, high-resolution stream network dataset delineating stream network channels at 90m spatial resolution. The dataset comprises 1.6 million drainage basins, 726 million stream segments and

sub-catchments, along with 42 stream-topographical and -topological variables. The sub-catchments correspond to the stream segments as they share a unique ID. *Environment90m* includes 45 variables related to topography and hydrography, 19 climate variables for the observation period of 1981-2010, as well as projections for 2041-2070 and 2071-2100 under the Shared Socioeconomic Pathways (SSPs) 1.26, 3.70 and 5.85, and three global circulation models (UKESM, MPI and IPSL). Moreover, *Environment90m* includes 22 land cover categories for the annual time-series data from 1992-2020. In addition, we provide

15 soil variables and information on aridity and modelled streamflow. Summary statistics (i.e., mean, min, max, range, sd) are provided for all continuous variables while for categorical data, the proportion of each category is calculated within each of the sub-catchments. The data is available at https://hydrography.org/environment90m.

To mobilize the *Environment90m* data integration into workflows, we provide two options: first, we have implemented custom functions within the *hydrographr* R-package Schürz et al. (2023) for batch downloading, processing and integrating the

data directly with the Hydrography90m data, as well as to perform custom data processing routines. Second, all *Environment90m* variables are also available within the GeoFRESH online platform (available at https://geofresh.org/, Domisch et al. (2024)), which allows a fast download of the variables for point locations anywhere in the world, both for the given sub-catchment where the points are located as well as their upstream catchment. Moreover, the data can also be aggregated over lakes, as well as single river-lake intersections (Tomiczek et al., 2024) given the functions in the *hydrographr* R-package. Users

can therefore retrieve the *Environment90m* information for each lake's upstream catchment area. We showcase the workflow in this exemplary lake vignette https://glowabio.github.io/hydrographr/articles/case_study_lake_workflow.html.

## 2  Environmental Data

The *Environment90m* dataset consists of tabular data describing the summary statistics of different environmental datasets which we calculated for each single sub-catchment of the Hydrography90m dataset (Amatulli et al. (2022)). The following

describes the underlying environmental data to derive the *Environment90m* dataset.



## 2.1 Stream network data

The Hydrography90m is a high-resolution (∼90 m) dataset delineating a global stream channel network Amatulli et al. (2022) and serves as the base of *Environment90m*. The calculation of Hydrography90m used the MERIT Hydro Digital Elevation Model at 3 arcsec (∼90 m at the Equator) Yamazaki et al. (2017). The main feature of Hydrography90m is the delineation of small headwater streams. In addition, this dataset includes a number of stream topographic and topological properties (see Table1). The summary statistics of the *Environment90m* dataset presented here are calculated for each of the 726 million unique sub-catchments of the *Hydrography90m*, which also serve as the spatial units in *Environment90m*.

Table 1: List of variables derived from the *Hydrography90m* dataset

| **Hydrography90m** (Amatulli et al., 2022) | | | | |
|---|---|---|---|---|
| spatial resolution | $90m^2$ | | | |
| temporal resolution | - | | | |
| time range | - | | | |
| | | | | |
| **Variable Type** | **Variable** | **Abbreviation** | **Unit** | **Description** |
| Flow | flow accumulation | accumulation | $km^2$ | Accumulated number of cells that drain through each cell |
| Stream slope | cell maximum curvature | slope_curv_max_dw_cel | $m^{-1}$ | Cell maximum curvature (between highest upstream cell, focal cell and downstream cell). $Scale = 10^6$ |
| | cell minimum curvature | slope_curv_min_dw_cel | $m^{-1}$ | Cell minimum curvature (between lowest upstream cell, focal cell and downstream cell). $Scale = 10^6$ |
| | cell elevation difference | slope_elv_dw_cel | $m$ | Cell elevation difference (between focal cell and downstream cell) |
| | cell gradient | slope_grad_dw_cel | | $Scale = 10^6$ |
| Stream distance | shortest distance to drainage divide | stream_dist_up_near | $m$ | Shortest upstream distance between focal grid cell and the nearest sub-catchment drainage divide |
| | longest distance to drainage divide | stream_dist_up_farth | $m$ | Longest upstream distance between focal grid cell and the nearest sub-catchment drainage divide |
| | nearest down stream stream grid cell | stream_dist_dw_near | $m$ | Distance between focal grid cell and its nearest down stream stream grid cell |
| | outlet grid cell in the network | outlet_dist_dw_basin | $m$ | Distance between focal grid cell and the outlet grid cell in the network |



**Table 1 – continued from previous page**

| Variable Type | Variable | Abbreviation | Unit | Description |
|---|---|---|---|---|
| | downstream stream node grid cell | outlet_dist_dw_scatch | $m$ | Distance between focal grid cell and the down stream stream node grid cell |
| | euclidean distance | stream_dist_proximity | $m$ | Euclidean distance between focal grid cell and the stream network |
| Elevation | shortest path | stream_diff_up_near | $m$ | Elevation difference of the shortest path from focal grid cell to the sub-catchment drainage divide |
| | longest path | stream_diff_up_farth | $m$ | Elevation difference of the longest path from focal grid cell to the sub-catchment drainage divide |
| | nearest downstream stream pixel | stream_diff_dw_farth | $m$ | Elevation difference between focal grid cell and its nearest downstream stream pixel |
| | outlet grid cell in the network | outlet_diff_dw_basin | $m$ | Elevation difference between focal grid cell and the outlet grid cell in the network |
| | downstream stream node grid cell | outlet_diff_dw_scatch | $m$ | Elevation difference between focal grid cell and the downstream stream node grid cell |
| Segment properties | segment downstream mean gradient | channel_grad_dw_seg | | Segment downstream mean gradient (between focal cell and the node/outlet) |
| | segment upstream mean gradient | channel_grad_up_seg | | Segment upstream mean gradient (between focal cell and the init/node) |
| | cell upstream gradient | channel_grad_up_cel | | Cell upstream gradient (between focal cell and next cell) |
| | cell stream course curvature | channel_curv_cel | | Cell stream course curvature (focal cell) |
| | segment downstream elevation difference | channel_elv_dw_seg | | Segment downstream elevation difference (between focal cell and the node/outlet) |
| | segment upstream elevation difference | channel_elv_up_seg | | Segment upstream elevation difference (between focal cell and the init/node) |
| | cell upstream elevation difference | channel_elv_up_cel | | Cell upstream elevation difference (between focal cell and next cell) |
| | cell downstream elevation difference | channel_elv_dw_cel | | Cell downstream elevation difference (between focal cell and next cell) |





**Table 1 – continued from previous page**

| Variable Type | Variable | Abbreviation | Unit | Description |
|---|---|---|---|---|
| | segment downstream distance | channel_dist_dw_seg | | Segment downstream distance (between focal cell and the node/outlet) |
| | segment upstream distance | channel_dist_up_seg | | Segment upstream distance (between focal cell and the init/node) |
| | cell upstream distance | channel_dist_up_cel | | Cell upstream distance (between focal cell and next cell) |
| Stream order | Strahler's stream order | order_strahler | | |
| | Shreve's stream magnitude | order_shreve | | |
| | Horton's stream order | order_horton | | |
| | Hack's stream order | order_hack | | |
| | Topological dimension of streams | order_topo | | |
| Stream reach | Length of the stream reach | length | $m$ | Length of the stream reach |
| | Straight length | stright | $m$ | Length of the stream as straight line |
| | Sinusoid of the stream reach | sinosoid | | Fractal dimension: stream length/straight stream length |
| | Accumulated length | cum_length | $m$ | Length of stream from source |
| | Distance to outlet | out_dist | $m$ | Distance of current stream init from outlet |
| | Source elevation | source_elev | $m$ | Elevation of stream init |
| | Outlet elevation | outlet_elev | $m$ | Elevation of stream outlet |
| | Elevation drop | elev_drop | $m$ | Difference between source_elev and outlet_elev + drop outlet |
| | Outlet drop | out_drop | $m$ | Drop at the outlet of the stream |
| | Gradient | gradient | $m$ | Mean gradient of the sub-catchment (downstream elevation difference divided by distance) |
| Flow index | Stream power index | spi | | Measure of the erosive power of flowing water (Moore et al. (1991)) |
| | Sediment transportation index | sti | | Metric describing the erosion and deposition of sediments (Mojaddadi et al. (2017)) |
| | Compound topographic index | cti | | A steady state wetness index, also known as topographic wetness index (TWI) (Beven and Kirkby (1979)) |



**Table 1 – continued from previous page**

| Variable Type | Variable | Abbreviation | Unit | Description |
|---|---|---|---|---|
| Stream connectivity | Connectivity | connections | | attribute table with the sub-catchment id of the next stream segment (downstream: *next_stream*), and two or more contributing streams (upstream: *prev_stream*) |

## 2.2 Climate

We derived high-resolution climate information from the Chelsa v2.1 dataset available at https://chelsa-climate.org/ Karger et al. (2017a, 2021). We used 19 bioclimatic variables (bio 1 to 19) at 30-arc-sec (ca. 1 km) resolution (Table 2) for 30-year averages of temperature and precipitation. We aggregated the data for three time ranges: from 1981 to 2010, corresponding to observational data, and future projections for the years 2041 to 2070, as well as 2071 to 2100. For each future projection, we used the combination of three general circulation models (GCMs) (i.e., MPI, UKESM, IPSL) and three shared socioeconomic
pathways (SSP1-RCP2.6, SSP3-RCP7, and SSP5-RCP8.5; Ebi et al. (2014); O'Neill et al. (2017)).

Table 2: List of variables derived from the CHELSA dataset

| **Climatologies at high resolution for the earth's land surface areas. CHELSA v2.1** (Karger et al., 2017b) | |
|---|---|
| spatial resolution | $1km^2$ |
| temporal resolution | Long Term Annual Average |
| time range | 1981-2010, 2041-2070, 2071-2100 |
| circulation models | ipsl-cm6a-lr, mpi-esm1-2-hr, ukesm1-0-ll |
| shared socioeconomic pathways | ssp126, ssp370, spp586 |

| Variable Type | Variable | Abbreviation | Unit | Description |
|---|---|---|---|---|
| Temperature | annual mean temperature | bio01 | ° C | Scale = 0.1, Offset = -273.15: Mean annual daily mean air temperatures averaged over 1 year |
| | Mean diurnal range | bio02 | ° C | Scale = 0.1: Mean diurnal range of temperatures averaged over 1 year |
| | Isothermality | bio03 | ° C | Scale = 0.1: Ratio of diurnal variation to annual variation in temperatures |
| | Temperature seasonality | bio04 | ° C/100 | Scale = 0.1: Standard deviation of the monthly mean temperatures |





**Table 2 – continued from previous page**

| Variable Type | Variable | Abbreviation | Unit | Description |
|---|---|---|---|---|
| | Max temperature of warmest month | bio05 | °C | Scale = 0.1, Offset = -273.15: The highest temperature of any monthly daily mean maximum temperature |
| | Min temperature of coldest month | bio06 | °C | Scale = 0.1, Offset = -273.15: The lowest temperature of any monthly daily mean minimum temperature |
| | Temperature annual range | bio07 | °C | Scale = 0.1: The difference between the Maximum Temperature of Warmest month and the Minimum Temperature of Coldest month |
| | Mean temperature of wettest quarter | bio08 | °C | Scale = 0.1, Offset = -273.15: The wettest quarter of the year is determined (to the nearest month) |
| | Mean temperature of driest quarter | bio09 | °C | Scale = 0.1, Offset = -273.15: The driest quarter of the year is determined (to the nearest month) |
| | Mean Temperature of warmest Quarter | bio10 | °C | Scale = 0.1, Offset = -273.15: The warmest quarter of the year is determined (to the nearest month) |
| | Mean Temperature of coldest Quarter | bio11 | °C | Scale = 0.1, Offset = -273.15: The coldest quarter of the year is determined (to the nearest month) |
| Precipitation | annual precipitation | bio12 | $kg/m^2$ | Scale = 0.1: Accumulated precipitation amount over 1 year |
| | Precipitation of wettest month | bio13 | $kg/m^2$ | Scale = 0.1: The precipitation amount of the wettest month |
| | Precipitation of driest month | bio14 | $kg/m^2$ | Scale = 0.1: The precipitation amount of the driest month |
| | Precipitation seasonality | bio15 | $kg/m^2$ | Scale = 0.1: The Coefficient of Variation is the standard deviation of the monthly precipitation estimates expressed as a percentage of the mean of those estimates (i.e. the annual mean) |
| | Precipitation of wettest quarter | bio16 | $kg/m^2$ | Scale = 0.1: The wettest quarter of the year is determined (to the nearest month) |
| | Precipitation of driest quarter | bio17 | $kg/m^2$ | Scale = 0.1: The driest quarter of the year is determined (to the nearest month) |
| | Precipitation of warmest quarter | bio18 | $kg/m^2$ | Scale = 0.1: The warmest quarter of the year is determined (to the nearest month) |





**Table 2 – continued from previous page**

| Variable Type | Variable | Abbreviation | Unit | Description |
|---|---|---|---|---|
| | Precipitation of coldest quarter | bio19 | $kg/m^2$ | Scale = 0.1: The coldest quarter of the year is determined (to the nearest month) |

## 2.3 Land cover

For land use data, we aggregated the consistent global land cover maps of the Land Cover European Space Agency (ESA) Climate Change Initiative (CCI) project into 22 categories from the original 37 ESA category level 2 land cover classes at a
spatial resolution of 300m CCI (2017) (Table 3). The annual data are available for the years 1992 to 2020.





**Table 3.** List of variables (i.e., land cover categories) derived from the ESA land cover maps

**Consistent global land cover maps: ESA CCI land cover** (ESA, 2017)

| | |
|---|---|
| spatial resolution | $300m^2$ |
| temporal resolution | Annual |
| time range | 1992-2020 |

| Variable Type | Variable | Abbreviation | Unit | Description |
|---|---|---|---|---|
| Land Cover | Cropland | c10 | proportion | Cropland, rainfed (10, 11, 12) |
| | Cropland | c20 | proportion | Cropland, irrigated or post-flooding (20) |
| | Cropland/natural vegetation | c30 | proportion | Mosaic cropland (>50%) - natural vegetation (tree, shrub, herbaceous cover) (<50%) (30) |
| | Natural vegetation / cropland | c40 | proportion | Mosaic natural vegetation (tree, shrub, herbaceous cover) (>50%) / cropland (< 50%) (40) |
| | Tree cover, broadleaved, evergreen | c50 | proportion | Tree cover, broadleaved, evergreen, closed to open (>15%) (50) |
| | Tree cover, broadleaved, deciduous | c60 | proportion | Tree cover, broadleaved, deciduos, closed to open (>15%) (60 61 62) |
| | Tree cover, needleleaved, evergreen | c70 | proportion | Tree cover, needleleaved, evergreen, closed to open (>15%) (70 71 72) |
| | Tree cover, needleleaved, deciduous | c80 | proportion | Tree cover, needleleaved, deciduous, closed to open (>15%) (80 81 82) |
| | Tree cover, mixed leaf type | c90 | proportion | Tree cover, mixed leaf type (broadleaved and needle-leaved) (90) |
| | Tree and shrub | c100 | proportion | Mosaic tree and shrub (>50%) / herbaceous cover (<50%) (100) |
| | Herbaceous/tree and shrub | c110 | proportion | Mosaic herbaceous cover (>50%) / tree and shrub (<50%) 110 |
| | Shrubland | c120 | proportion | Shrubland (120 121 122) |
| | Grassland | c130 | proportion | Grassland (130) |
| | Lichens, mosses | c140 | proportion | Lichens, mosses (140) |
| | Sparse vegetation | c150 | proportion | Sparse vegetation (tree, shrub, herbaceous cover) (<15%) (150, 151, 152,153) |
| | Tree cover, flooded, fresh/brackish water | c160 | proportion | Tree cover, flooded, fresh or brackish water (160) |
| | Tree cover, flooded, saline water | c170 | proportion | Tree cover, flooded, saline water (170) |
| | Shrub or herbaceous | c180 | proportion | Shrub or herbaceous cover, flooded, fresh - saline - brackish water (180) |
| | Urban areas | c190 | proportion | Urban areas (190) |
| | Bare areas | c200 | proportion | Bare areas (200 201 202) |
| | Water bodies | c210 | proportion | Water bodies (210) |
| | Snow and ice | c220 | proportion | Permanent snow and ice (220) |



## 2.4 Soil

The 15 soil variables were sourced from the global gridded soil information dataset, SoilGrids250 v2.0 (Hengl et al., 2017). This dataset represents global chemical and physical soil properties (Table 4). Each of the variables was originally provided at six standard depths (with the exception of depth to bedrock and soil organic carbon content) and at a spatial resolution of 250 m. To integrate all available soil depths (up to 200 cm), we calculated the weighted average for each soil property originally measured at different depths (Hengl et al. (2017)).

**Table 4.** List of variables derived from the SOILGRID database

**SoilGrids: global gridded soil information (Hengl et al., 2017)**

| spatial resolution | $250m^2$ |
| --- | --- |
| temporal resolution | - |
| time range | - |

| Variable Type | Variable | Abbreviation | Unit | Description |
| --- | --- | --- | --- | --- |
| Soil | Derived saturated water content | awcts | | |
| | Clay content | clyppt | % | |
| | Sand content | sndppt | % | |
| | Silt content | sltppt | % | |
| | Derived available soil water capacity | wwp | | |
| | Soil organic carbon content | orcdrc | g/kg | |
| | Soil ph | phihox | pH | Soil pH x 10 in H2O |
| | Bulk density | bldfie | $Kg/m^3$ | |
| | Cation exchange capacity | cecsol | cmolc/kg | |
| | Coarse fragments volumetric | crfvol | % | |
| | Grade of a sub-soil being acid | acdwrb | pH | Grade of a sub-soil being acid e.g. having a pH < 5 and low BS |
| | Depth to bedrock (r horizon) up to 200 cm | bdricm | cm | |
| | Probability of occurrence of r horizon | bdrlog | % | |
| | Cumulative probability of organic soil | histpr | | Cumulative probability of organic soil based on the TAXOUSDA and TAXNWRB |
| | Sodic soil grade | slgwrb | pH | Sodic soil grade based on WRB soil types and soil pH |





## 2.5 Elevation

To represent the elevation variable we used the 90 m resolution Multi-Error-Removed Improved-Terrain Digital Elevation Model (MERIT DEM) (Yamazaki et al., 2017) (Table 5). The error removal procedures applied to this dataset have improved
its vertical accuracy. This dataset was also used as the basis for the creation of the *Hydrography90m* dataset (Amatulli et al., 2022).

**Table 5.** List of variables derived from the MERID DEM

| MERIT DEM: Multi-Error-Removed Improved-Terrain DEM v1.0.3 (Yamazaki et al., 2017) | | | | |
|---|---|---|---|---|
| spatial resolution | $90m^2$ | | | |
| **Variable Type** | **Variable** | **Abbreviation** | **Unit** | **Description** |
| Elevation | elevation | elev | $m$ | The MERIT DEM represents elevation in meters |

## 2.6 Stream flow

For stream (water) flow we used the FLO1K dataset which comprises the mean, maximum and minimum annual flow for each year in the period 1960–2015, provided as spatially continuous gridded layers at 30 arc-seconds (ca. 1 km) (Barbarossa et al.
(2018)) (Table 6). For *Environment90m*, we only used the data from 1980-2010 and averaged them across this time frame, to match the CHELSA observed climate dataset 2.2.

**Table 6.** List of variables derived from the FLO1K streamflow dataset

| FLO1K, global maps of mean, maximum and minimum annual streamflow (Barbarossa et al., 2018) | | | | |
|---|---|---|---|---|
| spatial resolution | $1km^2$ | | | |
| temporal resolution | Long term annual average | | | |
| time range | 1980-2010 | | | |
| **Variable Type** | **Variable** | **Abbreviation** | **Unit** | **Description** |
| Flow | streamflow | flo1k | $m^3/s$ | The long-term mean annual flow represents the average of the year-specific FLO1K maps for mean Annual Flow over the period 1980-2010 |

## 2.7 Global Aridity Index and Potential Evapotranspiration

This dataset provides high-resolution (30 arc-seconds, ca. 1 km) global raster data on evapotranspiration processes and rainfall deficit for potential vegetation growth. Global Aridity and Potential Evapotranspiration are both modeled using data available
from WorldClim Global Climate Data. The data is available for the 1970-2000 period (Zomer and Trabucco (2022))(Table 7).





**Table 7.** List of variables derived from the Global Aridity and Evapotranspiration dataset

| Global Aridity Index and Potential Evapotranspiration Climate Database v3 (Zomer and Trabucco, 2022) | | | |
|---|---|---|---|
| spatial resolution | $1km^2$ | | |
| temporal resolution | Long Term Average | | |
| time range | 1970-2000 | | |
| **Variable Type** | **Variable** | **Abbreviation** | **Unit** | **Description** |
| Evapotranspiration | gevapt | $mm$ | Potential Evapo-Transpiration (ET0) based upon implementation of the FAO-56 Penman-Monteith Reference Evapotranspiration (ET0) equation. |
| Aridity index | garid | | Ratio between precipitation and ET0. Values reported have been multiplied by a factor of 10.000 |

## 3 Calculations

For all sub-catchments available in the *Hydrography90m* dataset, we calculated different summary statistics for each of the environmental datasets described in section 2, and the resultant tables have been made available in different formats (Figure 1).

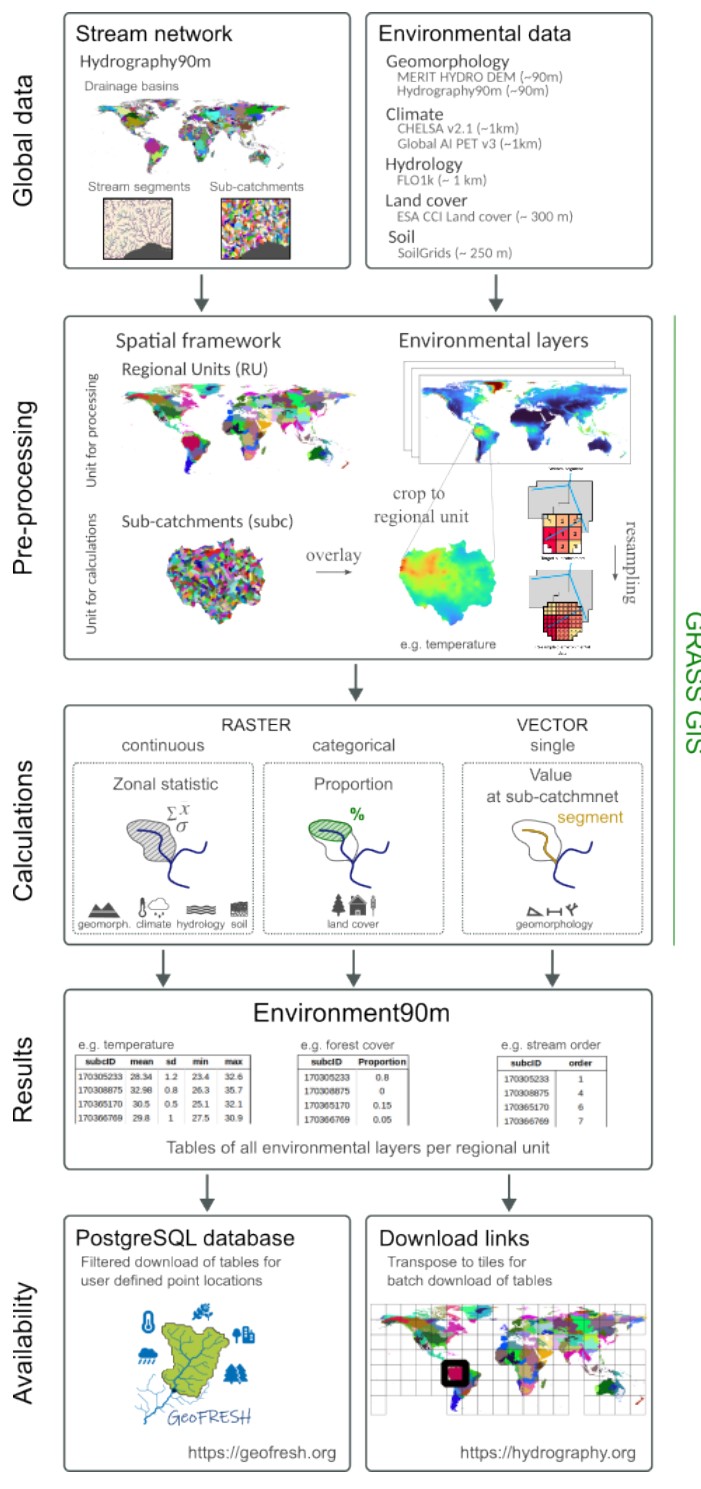

**Figure 1.** Workflow for the calculations of the *Environment90m* dataset

The procedure started by creating a working session in GRASS GIS for each of the 166 Regional Units (RUs) defined in
Amatulli et al. (2022), (see their Fig.7). Regional units are groups of one large or several small entire drainage basins, ensuring
that the whole area of the sub-catchments is included in the RU and to improve efficiency for computational calculations. The
GRASS GIS session initialization was made with the raster file of each RU containing the sub-catchments, which automatically
defined the geographical extent (i.e., the bounding box of the raster file) and the resolution ($90m^2$) as the default settings of the
session. Also, the default coordinate reference system was set to the World Geodetic System 1984 (WGS 84) with coordinates
expressed as latitude and longitude and defined by the EPSG:4236.

The raster files representing the environmental variables were then read into each of the GRASS GIS sessions, where the
software automatically cropped and resampled the original datasets to the same extent and resolution as the default settings. In
all our cases, the original raster files had the same or a lower resolution as 90 m given the *Hydrography90m* dataset. In case
the environmental data had a lower resolution, e.g. CHELSA climate at a native 30-arc-sec (1 km$^2$) resolution, these grid cells
are resampled to 90 m without interpolation, i.e. all new 90 m cells are assigned the same value as 1 km cells if they overlap
GRASS Development Team (2024).(Figure 2).

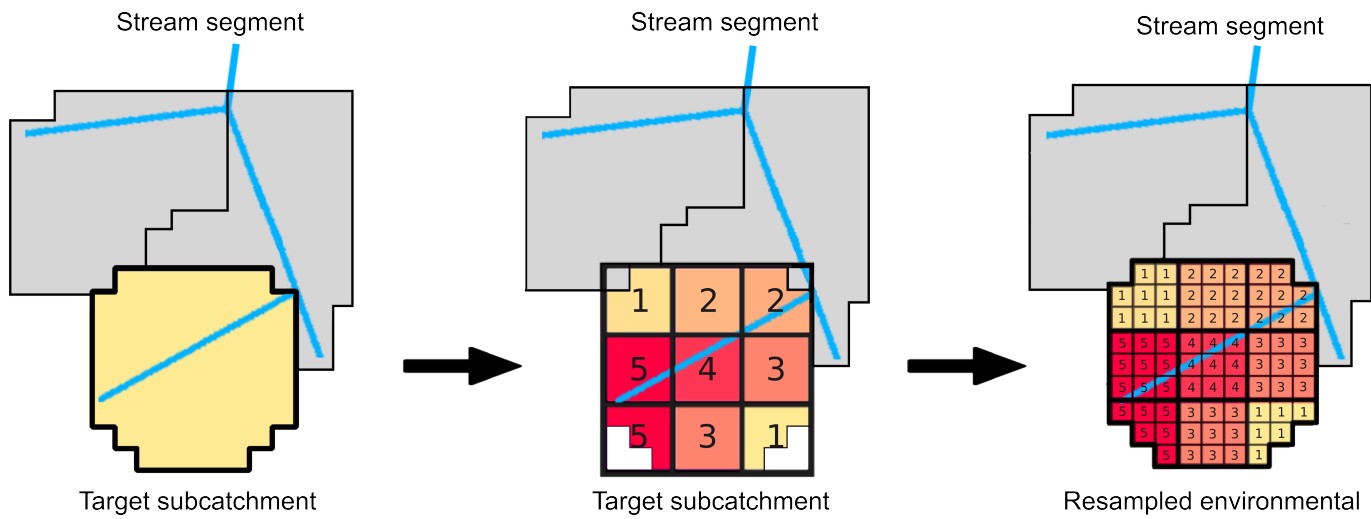

**Figure 2.** Automatic resample procedure (low to high resolution) when reading raster files into the default settings of a GRASS GIS session

Depending on the properties of each environmental dataset, a selection between three possible summary statistics was chosen
to calculate the output tables. The three categories were as follows:

1. Zonal statistics: calculation of the mean, standard deviation, range, minimum, and maximum of the environmental layer
145        within each sub-catchment. For this category, environmental layers representing continuous values (e.g. temperature)
were used. The calculations were done within the GRASS GIS environment using the `r.univar` function.





2. Proportion: the proportion of the variable (i.e., variable categories) in each sub-catchment. Categorical data were used here, specifically, the land cover. The calculations were done within the GRASS GIS environment by dividing the number of pixels of the target category in each sub-catchment with the total number of pixels within the same sub-catchment.

3. Value at sub-catchment: the *Hydrography90m* dataset provides a vector file with a list of attributes for every single stream segment of the global network. Since every sub-catchment share a unique ID with each stream segment, the value assigned to the sub-catchment corresponds to the value of the different attributes in the stream segment vector file (Amatulli et al. (2022). Examples of these attributes are e.g. stream length, or Strahler stream order.

The initial set of tables for all environmental variables covered the entire sub-catchments within each RU. These tables have been integrated into a PostgreSQL database as a backbone for the GeoFRESH online platform (available at www.geofresh.org, (Domisch et al., 2024)) where users can interactively retrieve the data for any location of interest. In addition, all tables follow the same tiling scheme as in the *Hydrography90m* dataset, such that the *Environment90m* and Hydrograhy90m datasets are compatible regarding the downloading and processing functionalities of the hydrographr R-package Schürz et al. (2023). All calculations were processed in parallel using the High Performance Computing (HPC) facility at Yale University.

## 4   Case study workflow

The *Environment90m* database is especially suited for freshwater biogeographic analyses, including predictive modelling of freshwater species distributions. This task usually requires range-wide spatial data and environmental data, which at high resolution, quickly becomes massive. To facilitate the acquisition and manipulation of the large tables of *Environment90m*, and to enable a fluent integration with the *Hydrography90m* network, we have developed additional functionalities in the hydrographr R-package (Schürz et al., 2023). Although manipulating data frames in R is usually easy, the added value of the new functionalities is to deal with the size of the tables, especially at large geographical extents, and to process and e.g. subset the large tables efficiently using R-commands, however using open-source third-party command-line tools without actually reading the data into R (which is one of the main features of the hydrographr package). The following case study illustrates a workflow example to create a map with the predicted probabilities of occurrence of the Danube streber fish *Zingel streber*, a species of freshwater ray-finned fish in the family Percidae (Figure 3). We provide the workflow at https://glowabio.github.io/hydrographr/articles/case_study_Danube.html.

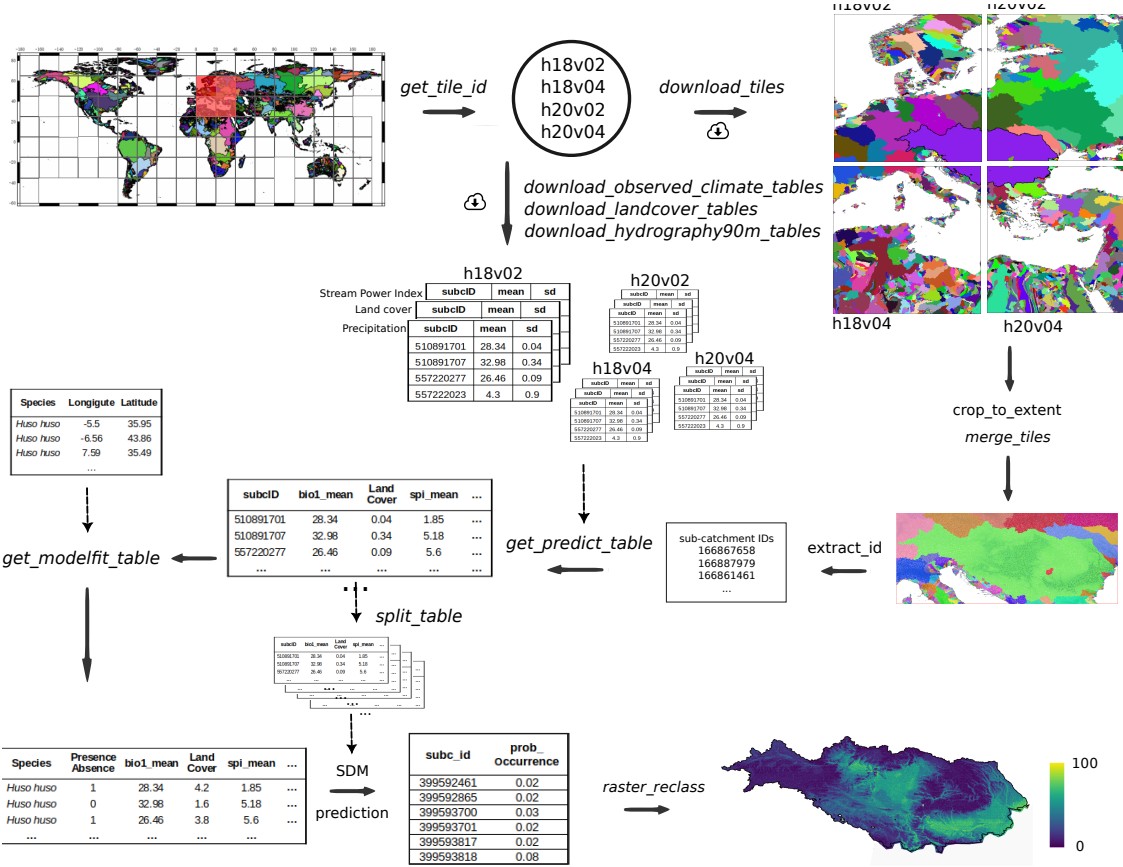

**Figure 3.** Case study workflow: create a map with the predicted probabilities of occurrence of the Danube streber fish *Zingel streber*

The first step is to identify the 20° x 20° tiles that overlap with a bounding box polygon of the Danube River basin by applying the function `get_tile_id`. Since the units of analysis to model the distribution of the species are the sub-catchments, we need to download the raster files of sub-catchments for each of the tiles using the `download_tiles` function, crop each
tile to the extension of interest with the function `crop_to_extent` and merge the pieces of each tile with the function `merge_tiles` to obtain a final raster file of sub-catchments for the bounding box of the Danube basin.

A parallel task is to download the corresponding tables for each tile of the selected environmental variables. There are a number of functions dedicated to download each of the available datasets (e.g. `download_landcover_tables`). The tables will be downloaded to disk and from here, they can be subset and merged, for example with the sub-catchment IDs
only present in the area of interest, in our case the Danube basin. This processing is done internally with the new function `get_predict_table` which uses as arguments (i) the path on disk where the downloaded tables are located, and (ii) the list of sub-catchment IDs which have been previously identified, using the function `extract_id` on the sub-catchment raster file of the area of interest. Here, either all or only a subset of the aggregation statistics (e.g., mean, range) can be selected.





The output is a large table (i.e., the so-called range-wide "prediction table") with all the sub-catchments of the area of interest

and the values of the selected environmental variables. This table is still on disk, and for initial screening, a subset of it can be loaded into R to run exploratory analyses or correlation analyses to make a selection of uncorrelated variables if the purpose is to e.g. quantify species ecological niches.

The species distribution modelling requires as input a table relating the species occurrence locations with the environmental data at those locations (i.e., the so-called "model fit table"). The function `get_modelfit_table` creates this table by

combining (i) a table of species geographic locations (i.e., coordinates), (ii) the previously created range-wide prediction table, and (iii) the raster of sub-catchments generated during the first steps of the workflow. The model fit table should contain the species occurrences and absences (or pseudo-absences) and their associated environmental values. The user can provide the occurrences and self-created pseudo-absences together. Alternatively, the function offers the possibility to create a user defined number of random pseudo-absences.

The model fit table can be imported into R, where any modelling technique (e.g., Random Forest) can be applied to estimate the ecological niche of the species and predict the probability of occurrence of the species in the area of interest. The prediction consists of a table with each sub-catchment id and its corresponding probability of occurrence value. This table can then be used with the `raster_reclass` function to reclassify the original sub-catchment raster file to create a new probability of occurrence raster file.

## 200  5   Conclusions

The availability of globally standardized environmental data that addresses the hydrographic network structure enables comparative studies across regions, and therefore facilitates large-scale biogeographical analyses in the freshwater realm. This makes *Environment90m*, which corresponds to the high spatial resolution of *Hydrography90m* stream network and the delineation of headwater streams, particularly valuable for global-scale freshwater biodiversity research, as shown in a number of

applications. For instance, the dataset was used in the Global EPTO Database (Grigoropoulou et al., 2023), where each insect occurrence record was linked to its corresponding sub-catchment and annotated with variables to facilitate spatial biodiversity analyses. Similarly, a recent study focusing on the Guineo-Congolian region, a biodiversity hotspot in the Afrotropics, integrated stream network attributes of the *Hydrography90m* derived from *Environment90m* with macroinvertebrate occurrence records spanning 2,890 sub-catchments and stream orders 1–12, enabling biogeographic analyses in a previously understudied

region. An application of the dataset in large-scale biodiversity assessments is demonstrated by (Haase et al., 2023), Haase et al., (2023), who used the dataset to analyse freshwater invertebrate diversity temporal trends across Europe. To identify environmental predictors that might drive these trends, the study used topographic, climatic and land-use variables aggregated at the sub-catchment level, drawn from *Environment90m*. In addition, *Environment90m* is the backbone of a recent global study on ecological niche breadths of aquatic insect genera worldwide, where a suite of environmental predictors, such as mean stream

slope gradient, stream length, bioclimatic variables and soil characteristics, was extracted per sub-catchment. These variables formed the basis for characterising genus-level niches using the Climate-niche factor analysis (CNFA) and assessing patterns





of aquatic insect niche breadth across freshwater insect assemblages (Grigoropoulou et al., in review). Moreover, a study from the University of California focusing on how the extent of permafrost sets the drainage density in the Arctic (Vecchio et al. (2024)) used the *Hydrography90m* dataset, derived from *Environment90m*, to calculate the drainage density in arctic water-

sheds between 23.5° and 90 °N latitude. Other authors have subsequently credited *Hydrography90m*, part of *Environment90m*, as a good source to derive and map headwaters and analyze streamflow dynamics, after using the dataset in their study on advancing the science for global water protection (Golden et al. (2025)).

Such applications demonstrate the value of the *Environment90m* dataset for freshwater biodiversity research worldwide, where globally standardised data accounting for the network structure are needed. A particular strength of the *Environment90m*

data and tools presented here are that they are seamlessly integrated with the existing *Hydrography90m* dataset, the hydrographr R-package and workflows to create and advance novel freshwater studies (see vignette examples). We highlight that the hydrographr R-package allows also to calculate custom variables across a given study area. Moreover, the GeoFRESH online platform, available at https://geofresh.org/ offers an additional avenue of retrieving *Environment90m* data. The graphical user interface allows to upload point coordinates to the portal, move (or "snap") the coordinates to the *Hydrography90m* stream

network, and annotate the coordinates with *Environment90m* variables Domisch et al. (2024).

We acknowledge that *Environment90m* focuses mainly on lotic habitats. To extend the data usage also to lentic habitats, we offer the possibility to extract *Environment90m* data not only for rivers but also across lakes and their contributing catchments. For this purpose, we have created new functions that are available in the hydrographr R package. Specifically, they allow identifying the location of a lake within the *Hydrography90m* stream network, and to extract the environmental variables

across the upstream catchment area for any lake connected to the network (currently pre-processed for the HydroLAKES dataset (Messager et al., 2016), though the functionality is generic for any lake dataset). For instance, by using the land cover data time series in *Environment90m*, it is possible to quantify the annual land cover changes in the catchment area for lakes of interest.

Taken together, we expect that *Environment90m* offers a unique possibility in analysing the environmental contingencies of

freshwater habitats at high spatial resolution. Moreover, the dataset supports biogeographical analyses of freshwater habitats and biodiversity, and contributes towards the recent freshwater biodiversity conservation targets by providing a solid and globally standardized baseline of high-resolution environmental information.

*Code availability.*

We provide all code for creating the *Environment90m* dataset at https://github.com/glowabio/environment90m

*Data availability.*





The metadata of the *Environment90m* dataset is stored at https://fred.igb-berlin.de/data/package/995 (García Márquez et al., 2025).

The *Environment90m* data can be obtained from the following sources:

- The primary *Environment90m* data is available as zipped .csv-tables. The data comes in 20° x 20° tiles, covering the same geographic extent and structure as the *Hydrography90m* dataset. These tiles can be interactively downloaded from https://hydrography.org/environment90m.

- We recommend downloading and attaching the tables to the *Hydrography90m* stream network using the *hydrographr* R-package (Schürz et al. (2023). We provide example code at https://glowabio.github.io/hydrographr/articles/case_study_Danube.html

- For single point occurrences (i.e. coordinates), we offer the possibility to upload these to the GeoFRESH online platform Domisch et al. (2024), available at https://geofresh.org/, and extract and download the *Environment90m* data either for the focal sub-catchment, or the aggregated data for the upstream contributing area.

*Author contributions.*

JGM and SD designed the study. JGM developed and implemented the workflow and processing chain in the Yale-HPC
to compute the *Environment90m* data. MB processed the data for the download and added the download functionality to the https://hydrography.org website. VB added the *Environment90m* data to the PostgreSQL database. YTC, VB and AG added the download functionality to the GeoFRESH online portal. JGM, MB, AG, MS, TT and YTC wrote the functions to download and process the data in the hydrographr R-package. All authors discussed the results, and all authors contributed to the writing of the manuscript.

*Competing interests.*

The contact author has declared that none of the authors has any competing interests.

*Acknowledgements.* We acknowledge funding through the Leibniz Competition project "Global freshwater biodiversity, biogeography and conservation (https://glowabio.org/; J45/2018). This work received also funding by the German Research Foundation (NFDI4Earth, DFG project no. 460036893, https://www.nfdi4earth.de; and NFDI4Biodiversity, DFG project number 442032008, https://www.nfdi4biodiversity.org),
and DFG reference number DO 1880/6-1, project number 533943718. We also acknowledge funding by the European Commission's Horizon Europe Research and Innovation programme under grant agreement numbers 101094434 (AquaINFRA), 101059264 (SOS-Water - Towards defining a safe operating space for the entire water resources in a changing climate and society), 101093985 (DANUBE4all) and by the German Federal Ministry of Education and Research (BMBF grant agreement number 033W034A). Yusdiel Torres-Cambas received funding by





the Alexander von Humboldt Foundation (Ref. 3.2-CUB-1212347-GF-P) and DFG (CRC RESIST, SFB 1439/1 2021–426547801). Vanessa

Bremerich acknowledges funding through the Leibniz Competition project "Freshwater Megafauna Futures" and DFG (CRC RESIST, SFB 1439/1 2021–426547801). Kristi Bego was funded by the German Academic Exchange Service (DAAD reference number 91902112). Finally, we thank the Yale Center for Research Computing for guidance and use of the research computing infrastructure.



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
