# Peer review of "Environment90m - globally standardized environmental variables for spatial freshwater biodiversity science at high spatial resolution"

_Earth System Science Data, 2025_

## Referee Comment (RC1)

**Review of the manuscript:**

**"Environment90m - globally standardized environmental variables for spatial freshwater biodiversity science at high spatial resolution"**

**Short summary**

In the data paper "Environment90m - globally standardized environmental variables for spatial freshwater biodiversity science at high spatial resolution", the authors present an aggregation of available global environmental datasets resampled for the sub-catchments of the Hydrography90m dataset. The variables include topography, hydrography, present and future climate variables, land cover, soil variables, aridity, and modelled streamflow.

In the **introduction**, the authors describe the need to protect freshwater biodiversity and habitats. To conduct globally standardised analysis and modelling studies, scientists require environmental data with a very high spatial resolution and global extent. The Environment90m dataset, based on the Hydrography90m stream network (Amatulli et al., 2022), should fill this gap.

In the **Environmental Data** section, each of the individual underlying datasets (all of which were published previously) is presented. An overview table is given for each dataset: Stream network data; climate; land cover; soil; elevation; stream flow; global aridity; and potential evapotranspiration.

In the **Calculations** section, the authors describe the dataset subsampling procedure. They also describe how the data is accessible to users.

The **Case study workflow** provides an example of how to use the dataset to predict the distribution of fish species in the Danube region.

The article ends with a **conclusion** that summarises applications of the dataset and potential possibilities to extract also data for lotic habitats.

**Main concerns**

After carefully reading the research article "Environment90m - globally standardized environmental variables for spatial freshwater biodiversity science at high spatial resolution", I see an immense value in the presented dataset. The data presented is not new itself, but the consistent aggregation and re-sampling on a global scale is of high value for further studies. Therefore, it will be of high interest to a huge number of users. The dataset is very well accessibly by tools provided to access the data via an R package (hydrographr), an online platform (GeoFresh), or direct downloads seem very helpful and are well documented with vignettes.

*Review - "*Environment90m - globally standardized environmental variables for spatial freshwater biodiversity science at high spatial resolution*"*

However, I have some suggestion about how to improve the structure und presentation of the dataset, which I will outline below:

**Structure & Presentation of the dataset**

The manuscript would benefit from a clearer and more consistent structure.

To improve the structure, I would suggest the following new sections:

- Consider creating a dedicated section titled *"Accessing the Data"*. Currently, information about data access is scattered across the Introduction, Calculation, and Case Study sections. Consolidating this content would improve readability and help users locate key information more easily. This section could also include a more detailed explanation of the custom function in the hydrography R package.
- Consider adding a separate section for *Applications* rather than embedding these points within the Conclusions. This would allow for a more focused discussion and better highlight the broader relevance of the dataset and tools.
- A more in-depth discussion of the dataset would strengthen the manuscript. For example:
  - What are the implications of the sub-sampling effort?
  - Could the dataset be expanded in future versions?
  - The absence of water chemistry parameters (e.g., nutrients) is notable, as these are key drivers of freshwater biodiversity. A brief discussion of this limitation and its potential impact on certain taxa would be valuable.

To improve the understandability of both the methods and the dataset, I suggest reordering the relevant sections: Start with the Calculations and then present the Environmental Data including the base datasets, selection criteria, pre-processing steps, and the final derived variables.

- Reason: Currently, the presentation of the dataset and the methods is interwoven, which leads to confusion. The original input datasets are introduced first, but the final dataset - including all derived summary statistics - is not clearly distinguished. This makes it difficult to grasp what is actually included in the published Environment90m dataset.
- Suggestion: I suggest reordering the section to first explain the resampling and calculation procedures (e.g., how zonal statistics, proportions, or other metrics were computed for the sub-catchments), and then introduce the Environment90m dataset (and it input data) as the result of these operations. This would help readers better understand the transformation from raw input data to processed variables. I also recommend updating the tables to include the available summary statistics for each variable. This would make the tables more informative and clearly reflect the contents of the final dataset.

Please check that all information is given within the respective section:

- For instance, the section titled *"Case Study Workflow"* begins with a discussion of the R package's usefulness, rather than introducing the case study itself.

*Review* - *"*Environment90m - globally standardized environmental variables for spatial freshwater biodiversity science at high spatial resolution*"*

**Language and readability**

There is potential to improve the readability of the manuscript. Some sentences are overly long and complex, which makes the text difficult to follow (see examples in the specific comments). Shorter sentences and a more active voice would enhance clarity and accessibility. The language check should be done also beyond the specifically mentioned sections in the specific comments.

The in-text citation style is inconsistent. Some references include brackets around the year, while others do not. In many cases, brackets around the entire citation are missing, which disrupts the flow of reading. Please ensure that the citation style is consistent throughout the manuscript and adheres to the journal's guidelines.

Figures and tables are often hard to read due to small font sizes, unclear terminology, or lack of explanatory notes (see specific comments).

**Specific comments**

**Title**

Two thoughts about the title of the paper:

- The current title emphasizes "spatial freshwater biodiversity science". I would encourage the authors to consider whether the dataset might also be relevant to other fields, such as hydrology, landscape ecology, or environmental modelling. If so, broadening the scope of the title could help reach a wider audience and better reflect the dataset's potential applications.
- The word *"spatial"* appears twice in the title, which may be redundant. Removing the first occurrence could help streamline the title without losing clarity.

**Introduction**

L19: "Freshwater biodiversity is among the terrestrial and marine realms most at risk." -> I do not properly understand the meaning of the sentence. It compares biodiversity (in freshwater) with realms (terrestrial and marine)? Please re-formulate more clearly. (e.g. "Freshwater biodiversity is considered to be more threatened than biodiversity in terrestrial and marine ecosystems.")

L19: Brackets around the citation are missing.

L19-23: The sentence is difficult to understand due to its length and complexity. I recommend streamlining and shortening it to improve readability. Breaking it into two or more sentences could help clarify the intended message.

L23-26: I suggest separating the question from the rest of the sentence with ":" – e.g. "(...) are required to answer the question: which areas should be prioritised for protection?"

L39-43: This is also a very long sentence. I can hardly understand it because of its length. Please streamline and shorten.

*Review - "*Environment90m - globally standardized environmental variables for spatial freshwater biodiversity science at high spatial resolution*"*

L45: "would lump the data" -> is this a standard term or colloquial language?

L50-51: The sentence *"Sub-catchments consist therefore of the natural units in freshwater ecosystems and allow encompassing also riparian areas and aquatic-terrestrial linkages."* is difficult to interpret. The term *"natural units"* is unclear — could the authors clarify what is meant by this? Does it refer to ecological boundaries, hydrological divisions, or something else?

Additionally, the word *"therefore"* is typically set off by commas when used in this context. A revision for clarity and grammatical correctness is recommended.

L57: Why is here a question mark?

L81: The following mentioned link to the exemplary lake vignette is not working: https://glowabio.github.io/hydrographr/articles/case_study_lake_workflow.html.

**Environmental Data**

L 83-85: When I was reading the paper for the first time I was confused about the content of the section, whether this section refers only to the input data available from Hydrography90 and other datasets or also includes the processed variables used in this paper. Therefore, I suggest changing the following sentence "The following describes the underlying environmental data to derive the Environment90m dataset." into "The following section describes the foundational environmental datasets used to generate the Environment90m dataset."

*General comment on spatial resolution*: I struggled to understand the spatial resolution descriptions throughout the manuscript. For example, if you refer to a resolution of "90 m" (L. 87), while in table 1 it is written as "90 m²". This creates confusion: If 90 m refers to the length of one pixel, then the actual area covered by a pixel would be 90 m × 90 m = 8100 m². This same issue appears with other datasets as well (e.g. land cover, soil). Please clarify whether the resolution values refer to pixel length or pixel area and ensure consistency across the manuscript and tables.

L87 ff: Be consistent in formatting the data sets in italics.

L91-92: Could you briefly explain how the 726 million sub-catchments can be identified and located? It would be helpful to know where users can find their IDs, locations, and areas, especially for working with the dataset.

L99-100: The sentence suggests that a "combination" of three SSPs and three GCMs was used to derive the climate variables. Could the authors clarify what this means in practice? Was the mean calculated across all combinations of SSPs and GCMs, resulting in a single averaged output? Or were individual outputs generated for each SSP-GCM pairing? Please describe it more precise. And is this a step done by the authors (and should be described in the Calculations) or is this an aggregation of the original dataset?

L 103-105: Please clarify whether the data refers to land use, land cover, or both. The terminology should be consistent throughout the manuscript to avoid confusion. It seems that a reference to the

data source is missing in this section. Including a citation or link to the original dataset would improve transparency and reproducibility. Additionally, it would be helpful to explain which categories were aggregated and how.

L110: The sentence mentions weighting based on soil depth, but it's not entirely clear how this was implemented. Was the variable weighted by the depth of each respective soil layer?

L118: Why are annual stream flow data not provided, along with mean, maximum, and minimum annual values? This type of information could be highly valuable for environmental modelling, especially when assessing long-term hydrological patterns or linking stream flow to land use changes. Since land use data is provided annually, aligning the temporal resolution of stream flow data would improve consistency and facilitate integrated analyses. I recommend including these annual metrics or explaining why they were omitted.

**Calculations**

L130: Please remove the comma.

L 140: Remove the period within the brackets.

L139-140: Could the authors clarify why interpolation was not applied? In many cases, interpolation can help fill gaps or smooth spatial data. A brief explanation of this decision would help readers understand the methodological choices and any limitations that result.

**Case study workflow**

The current presentation of the case study reads more like a methods tutorial than a full use case. If the intention is to present a complete use case, I suggest including:

- A short introduction to the ecological or scientific relevance.
- A summary of results.
- A brief discussion of those results.

If, however, the goal is to present only the workflow, then the link to the provided vignette is maybe sufficient, as it allows for code visibility and intermediate outputs.

**Tables**

The descriptions (column description) within the individual tables were not always understandable, some explanations of abbreviations of terms like "focal grid cell" or "Scale" are missing. See individual comments below.

*Table 1*
- Spatial resolution: Is the resolution really 90 m²? Based on my understanding, each pixel in Hydrography90m has a length of 90 meters, which would result in an area of 90 m × 90 m = 8100 m². Please clarify this.

- Scale = $10^6$: The meaning of this scale value is unclear. Does it refer to map scale, data normalization, or something else? A brief explanation would be helpful.
- Focal grid cell: The term is not clearly defined. I had assumed the data is aggregated at the sub-catchment level rather than at individual grid cells. Please clarify what is meant by "focal grid cell" in this context.

*Table 2*

- The meaning of Scale and Offset (e.g., "Scale = 0.1, Offset = -273.15") is not explained. Are these used for data transformation or unit conversion? If so, please provide a short note or example.
- "annual precipitation": Consider capitalizing the first letter for consistency with other entries.

*Table 3*

- What is the meaning of the numbers in the description?
- "Water bodies": Is there a possibility to differentiate between types of water bodies (e.g., lakes, rivers, streams)? This could add valuable ecological context.

*Table 4*

- The description is difficult to understand due to unexplained abbreviations. Please avoid using abbreviations unless they are defined in the table caption or footnotes.

*Table 5*

- temporal resolution and time range are missing

**Figures**

*Figure 1*

This is a visually engaging and informative overview figure. However, I recommend increasing its size for better readability (maybe some re-arrangements of the elements is needed). Currently, the font size is quite small, which makes it difficult to interpret some of the details.

*Figure 2*

I do not see that this figure adds new information beyond what is already presented in Figure 1.

*Figure 3*

This figure is also difficult to read due to its small size and dense content. Enlarging the figure and improving the layout or font size would help make it more accessible to readers.

**Supplementary material**

I appreciate the inclusion of multiple vignettes, tutorials, and external web resources linked in the paper. These additions significantly enhance the usability and accessibility of the dataset and tools, and they provide valuable guidance for potential users. Well done!

---

## Author Comment (AC1)

After carefully reading the research article "Environment90m - globally standardized environmental variables for spatial freshwater biodiversity science at high spatial resolution", I see an immense value in the presented dataset. The data presented is not new itself, but the consistent aggregation and re-sampling on a global scale is of high value for further studies. Therefore, it will be of high interest to a huge number of users. The dataset is very well accessibly by tools provided to access the data via an R package (hydrographr), an online platform (GeoFresh), or direct downloads seem very helpful and are well documented with Vignettes. However, I have some suggestion about how to improve the structure und presentation of the dataset and methods, as well as the language, which I will outline in the attached PDF.

We thank the reviewer for the valuable comments and constructive criticism. We took all suggestions into account during the revision of our manuscript.

Citation: https://doi.org/10.5194/essd-2025-399-RC1

Structure & Presentation of the dataset

The manuscript would benefit from a clearer and more consistent structure. To improve the structure, I would suggest the following new sections:

• Consider creating a dedicated section titled "Accessing the Data". Currently, information about data access is scattered across the Introduction, Calculation, and Case Study sections. Consolidating this content would improve readability and help users locate key information more easily. This section could also include a more detailed explanation of the custom function in the hydrography R package.

Thanks for the suggestion. We have created a new section 2.4 called "Accessing the Environment90m dataset" where we have compiled the accessibility information. Also, we added a table with the description of the new functions to download and process the large Environment90m tables.

• Consider adding a separate section for Applications rather than embedding these points within the Conclusions. This would allow for a more focused discussion and better highlight the broader relevance of the dataset and tools.

We have created the new section "Applications" (Section 5) and moved the related content from the Conclusions to this section.

• A more in-depth discussion of the dataset would strengthen the manuscript. For example:

  - What are the implications of the sub-sampling effort?

We extended the new Discussion section by mentioning the advantages of using the sub-catchments as units of analysis, in contrast to using grid cells, and therefore the

advantages of aggregating (sub-sampling) environmental data using these units, particularly in relation to the ecological relevance and the gain in new information potentially useful to describe and understand habitat and distribution patterns of freshwater ecosystems and biodiversity.

- Could the dataset be expanded in future versions?

Yes this is possible and we are actually looking into this at the moment (regarding the latest climate data, and updating the latest land cover data). Please see also the next comment.

- The absence of water chemistry parameters (e.g., nutrients) is notable, as these are key drivers of freshwater biodiversity. A brief discussion of this limitation and its potential impact on certain taxa would be valuable.

We added at the end of the Discussion section a paragraph referring to the additional data expected to be added to the Environment90m dataset in the near future. Also, key datasets, like chemistry parameters that did not fulfill the criteria to be included in this version. Water chemistry and nutrient data are usually not available range-wide, i.e. across the continuous surface, but mostly available as point data for certain basins and sub-catchments. The Environment90m dataset strives to make range-wide data, tailored towards freshwater biodiversity analyses, easily accessible, and we think that adding scattered geographic point data would not be in the scope of this manuscript.

To improve the understandability of both the methods and the dataset, I suggest reordering the relevant sections: Start with the Calculations and then present the Environmental Data including the base datasets, selection criteria, pre-processing steps, and the final derived variables.

• Reason: Currently, the presentation of the dataset and the methods is interwoven, which leads to confusion. The original input datasets are introduced first, but the final dataset - including all derived summary statistics - is not clearly distinguished. This makes it difficult to grasp what is actually included in the published Environment90m dataset.

• Suggestion: I suggest reordering the section to first explain the resampling and calculation procedures (e.g., how zonal statistics, proportions, or other metrics were computed for the sub-catchments), and then introduce the Environment90m dataset (and it input data) as the result of these operations. This would help readers better understand the transformation from raw input data to processed variables.

Thank you for the suggestion. We made some changes to improve the understandability of the procedures and creation of the Environment90m dataset. We started the methodology by providing the criteria to select the underlying environmental datasets, followed by how the calculations were done, then a description of each dataset with their respective tables and finally the data accessibility.

I also recommend updating the tables to include the available summary statistics for each variable. This would make the tables more informative and clearly reflect the contents of the final dataset.

We have added the information on the available statistics for each dataset in the header information of each table.

Please check that all information is given within the respective section:

• For instance, the section titled "Case Study Workflow" begins with a discussion of the R package's usefulness, rather than introducing the case study itself.

Thank you for spotting this. We moved the R package discussion to the same section where we introduce the new available functions of the package (i.e., section on accessibility - Section 5). In this way, the case study focuses exclusively on the particular workflow example.

Language and readability

There is potential to improve the readability of the manuscript. Some sentences are overly long and complex, which makes the text difficult to follow (see examples in the specific comments). Shorter sentences and a more active voice would enhance clarity and accessibility. The language check should be done also beyond the specifically mentioned sections in the specific comments.
Thank you for the suggestion. We went through the entire text and edited the long sentences.

The in-text citation style is inconsistent. Some references include brackets around the year, while others do not. In many cases, brackets around the entire citation are missing, which disrupts the flow of reading. Please ensure that the citation style is consistent throughout the manuscript and adheres to the journal's guidelines.
This was partly related to the Latex-libraries which we checked again. Citations are now consistent throughout the document.

Figures and tables are often hard to read due to small font sizes, unclear terminology, or lack of explanatory notes (see specific comments).
We edited the figures and tables and checked that the text is now easier to read.

Specific comments

Title

Two thoughts about the title of the paper:

• The current title emphasizes "spatial freshwater biodiversity science". I would encourage the authors to consider whether the dataset might also be relevant to other fields, such as hydrology, landscape ecology, or environmental modelling. If so, broadening the scope of the title could help reach a wider audience and better reflect the dataset's potential applications.

We fully understand the point of view, and we have discussed the title among the authors. We would like to keep the emphasis on freshwater, since this is the main scope of such "freshwaterised" data that takes the irregular spatial units in terms of sub-catchments along the stream network into account. We removed the term "biodiversity" to leave the scope open to different fields within the freshwater realm (e.g., hydrology, ecosystems, etc..).

• The word "spatial" appears twice in the title, which may be redundant. Removing the first occurrence could help streamline the title without losing clarity.

We removed the first "spatial"

Introduction

L19: "Freshwater biodiversity is among the terrestrial and marine realms most at risk." -> I do not properly understand the meaning of the sentence. It compares biodiversity (in freshwater) with realms (terrestrial and marine)? Please re-formulate more clearly. (e.g. "Freshwater biodiversity is considered to be more threatened than biodiversity in terrestrial and marine ecosystems.")

Changed to: "Freshwater ecosystems and biodiversity are considered to be more threatened than their terrestrial and marine counterparts".

L19: Brackets around the citation are missing.
All the citations have been properly standardized throughout the document.

L19-23: The sentence is difficult to understand due to its length and complexity. I recommend streamlining and shortening it to improve readability. Breaking it into two or more sentences could help clarify the intended message.

We re-wrote the sentence and splitted it into three parts.

L23-26: I suggest separating the question from the rest of the sentence with ":" – e.g. "(…) are required to answer the question: which areas should be prioritised for protection?"
Done

L39-43: This is also a very long sentence. I can hardly understand it because of its length. Please streamline and shorten.
Done.

L45: "would lump the data" -> is this a standard term or colloquial language?

We reduce the sentence by only saying that aggregating the data to larger units will challenge the attribution of environmental characteristics.

L50-51: The sentence "Sub-catchments consist therefore of the natural units in freshwater ecosystems and allow encompassing also riparian areas and aquatic-terrestrial linkages." is difficult to interpret. The term "natural units" is unclear — could the authors clarify what is meant by this? Does it refer to ecological boundaries, hydrological divisions, or something else? Additionally, the word "therefore" is typically set off by commas when used in this context. A revision for clarity and grammatical correctness is recommended.

We redefined sub-catchments as "sub-catchments can be considered as the smallest units with their own ecological and hydrological boundaries in freshwater ecosystems...".

L57: Why is here a question mark?

A reference was missing, which is now included (Latex inserts questionmarks when references are missing, and we did not spot this during prior submission).

L81: The following mentioned link to the exemplary lake vignette is not working: https://glowabio.github.io/hydrographr/articles/case_study_lake_workflow.html.
We changed the URL to https://glowabio.github.io/hydrographr/articles/case_study_lakes.html .

Environmental Data

L 83-85: When I was reading the paper for the first time I was confused about the content of the section, whether this section refers only to the input data available from Hydrography90 and other datasets or also includes the processed variables used in this paper. Therefore, I suggest changing the following sentence "The following describes the underlying environmental data to derive the Environment90m dataset." into "The following section describes the foundational environmental datasets used to generate the Environment90m dataset."

We have added the suggested sentence at the beginning of the description of the input datasets.

General comment on spatial resolution: I struggled to understand the spatial resolution descriptions throughout the manuscript. For example, if you refer to a resolution of "90 m" (L. 87), while in table 1 it is written as "90 m²". This creates confusion: If 90 m refers to the length of one pixel, then the actual area covered by a pixel would be 90 m × 90 m = 8100 m². This same issue appears with other datasets as well (e.g. land cover, soil). Please clarify whether the resolution values refer to pixel length or pixel area and ensure consistency across the manuscript and tables.

The resolution indeed refers to the length of the pixel. We have used , e.g. 90m consistently throughout the document.

L87 ff: Be consistent in formatting the data sets in italics.
They are now consistent throughout the document.

L91-92: Could you briefly explain how the 726 million sub-catchments can be identified and located? It would be helpful to know where users can find their IDs, locations, and areas, especially for working with the dataset.

The section of the case study workflow and the accompanying vignette will guide  users to the functions of the R package, which can be used to identify the location of specific sub-catchment ids, to subset the tables for the area of interest, and to integrate the tables to their own workflows.

L99-100: The sentence suggests that a "combination" of three SSPs and three GCMs was used to derive the climate variables. Could the authors clarify what this means in practice? Was the mean calculated across all combinations of SSPs and GCMs, resulting in a single averaged output? Or were individual outputs generated for each SSP-GCM pairing? Please describe itmore precise. And is this a step done by the authors (and should be described in the Calculations) or is this an aggregation of the original dataset?

We changed the sentence as follows to clarify the individual pairing of GCMs and SSPs: "We aggregated the reference period available as a long term annual average (1981-2010). Also, data for future projections (i.e., periods 2041-2070 and 2071-2100) were aggregated by selecting individual pairings of three global circulation models (GCMs) (i.e., mpi-esm1-2-hr, ukesm1-0-ll, ipsl-cm6a-lr) and three shared socioeconomic pathways (SSP1-RCP2.6, SSP3-RCP7, and SSP5-RCP8.5...".

L 103-105: Please clarify whether the data refers to land use, land cover, or both. The terminology should be consistent throughout the manuscript to avoid confusion. It seems that a reference to the data source is missing in this section. Including a citation or link to the original dataset would improve transparency and reproducibility. Additionally, it would be helpful to explain which categories were aggregated and how.

We are now consistent with "land cover" data. We also specified the reference to this dataset (CCI, 2017).
CCI, E. L. C.: Product user guide version 2.0, UCL-Geomatics: London, UK, 685, 2017.

The original categories of this dataset have been created in a hierarchical manner with different levels. Level 2 is meant to be used at regional scales and level 1 at the global scale. We adopted level 1 categories but added the following explanation in the caption of the table to clarify how the categories level 2 were aggregated: "...the numbers within parenthesis in the description make reference to the coding of the categories at level 2".

L110: The sentence mentions weighting based on soil depth, but it's not entirely clear how this was implemented. Was the variable weighted by the depth of each respective soil layer?

We follow the GlobalSoilMap specifications where averages over depth intervals, can be

derived by taking a weighted average of the predictions within the depth interval using numerical integration: 1 /(b−a)* 1/ 2 N ∑ k=1 (xk+1 −xk) (f(xk) + f(xk+1)). We added the reference to the specifications if the reader wants to know how the calculation was done. (Arrouays et al. 2014).

L118: Why are annual stream flow data not provided, along with mean, maximum, and minimum annual values? This type of information could be highly valuable for environmental modelling, especially when assessing long-term hydrological patterns or linking stream flow to land use changes. Since land use data is provided annually, aligning the temporal resolution of stream flow data would improve consistency and facilitate integrated analyses. I recommend including these annual metrics or explaining why they were omitted.

As explained in the text, we aim, initially, at matching the same long term average as the CHELSA (climate) dataset. We add in the conclusions, as part of future developments, that we will also provide the yearly average of stream flows.

Calculations

L130: Please remove the comma.
Removed.

L 140: Remove the period within the brackets.
Removed.

L139-140: Could the authors clarify why interpolation was not applied? In many cases, interpolation can help fill gaps or smooth spatial data. A brief explanation of this decision would help readers understand the methodological choices and any limitations that result.

We agree that interpolation techniques are very useful to fill gaps or smooth spatial data. However, we believe that interpolation is not required for the purpose of aggregating the original datasets to the sub-catchments. In fact, one criterion to select datasets to be included in the Environment90m dataset was that they have a high spatial resolution (1 km or higher) and have a globally standardized coverage. In general, all the selected variables have a complete overlap with the sub-catchments. Other potential variables, like the nitrogen deposition, which has a lower resolution and contains many gaps around the world were not included. We agree that data with these characteristics would definitely benefit from interpolation approaches to fill gaps.

Case study workflow

The current presentation of the case study reads more like a methods tutorial than a full use case. If the intention is to present a complete use case, I suggest including:
• A short introduction to the ecological or scientific relevance.
• A summary of results.
• A brief discussion of those results. If, however, the goal is to present only the workflow, then the link to the provided vignette is maybe sufficient, as it allows for code visibility and

intermediate outputs.

Indeed, the main purpose of the case study is to illustrate and provide users with an example workflow of how the large tables of the Environment90m dataset can be efficiently processed with the use of the new functions in the hydrographr R-package. We tried to summarize the case study in the paper to the users with the general workflow. The vignette is a more detailed option for users to use the code and evaluate temporal and final results.

Tables

The descriptions (column description) within the individual tables were not always understandable, some explanations of abbreviations of terms like "focal grid cell" or "Scale" are missing. See individual comments below.
Thank you for spotting these issues which we have now fixed.

Table 1

• Spatial resolution: Is the resolution really 90 m²? Based on my understanding, each pixel in Hydrography90m has a length of 90 meters, which would result in an area of 90 m × 90 m = 8100 m². Please clarify this.

Correct, the spatial resolution is 90m x 90m. We checked the text and corrected this.

• Scale = $10^6$ : The meaning of this scale value is unclear. Does it refer to map scale, data normalization, or something else? A brief explanation would be helpful.

It is a scale factor applied to the units of the variable. We explained the use of the scale factor within the caption of the table. We changed the position of the scale factor to the units (column) of the variable and removed it from the description column.

• Focal grid cell: The term is not clearly defined. I had assumed the data is aggregated at the sub-catchment level rather than at individual grid cells. Please clarify what is meant by "focal grid cell" in this context.

In the caption of the table we have defined the different terms used in the description of the variables. This description makes reference to the way each variable was calculated in its original form. The aggregation to the sub-catchment is a post-processing calculation done to create the summary tables available in the Environment90m dataset as explained in section 2.1.

Table 2

• The meaning of Scale and Offset (e.g., "Scale = 0.1, Offset = -273.15") is not explained. Are

these used for data transformation or unit conversion? If so, please provide a short note or Example.

We added to the caption the description of the scale and offsets factors and the equation to convert the raw values to a meaningful value, for example to convert the raw values to degree celcius for temperature related variables.

• "annual precipitation": Consider capitalizing the first letter for consistency with other entries.

Done.

Table 3

• What is the meaning of the numbers in the description?

The numbers in the description refer to the original values of the categories that were aggregated into the new categories for the environment90m dataset. We have added this explanation in the caption of Table 3.

• "Water bodies": Is there a possibility to differentiate between types of water bodies (e.g., lakes, rivers, streams)? This could add valuable ecological context.

Unfortunately not. The original land cover dataset only has one general category for water bodies.

Table 4

• The description is difficult to understand due to unexplained abbreviations. Please avoid using abbreviations unless they are defined in the table caption or footnotes.

The description of the soil variables is missing in most cases because the name of the variables are self explanatory. The abbreviations are only displayed as a guide for users to use this abbreviation to retrieve the data from the Environment90m dateset, for example by using the download functions in the hydrographr R-package. We added this explanation at the beginning of the underlying datasets section.

Table 5

• temporal resolution and time range are missing

Although these fields are empty for this dataset, we have added them for consistency with the other tables.

Figures

Figure 1

This is a visually engaging and informative overview figure. However, I recommend increasing its size for better readability (maybe some re-arrangements of the elements is needed). Currently, the font size is quite small, which makes it difficult to interpret some of the details.

We have re-designed the figure insets for more clarity and have increased the font size to improve its readability.

Figure 2
I do not see that this figure adds new information beyond what is already presented in Figure 1.

To avoid duplication, we have simplified the resampling illustration within Figure 1, such that Figs. 1 and 2 are now different. We would like to retain Figure 2 since the explanation of the resampling approach has been frequently requested when presenting the dataset at different venues.

Figure 3

This figure is also difficult to read due to its small size and dense content. Enlarging the figure and improving the layout or font size would help make it more accessible to readers.

We have enlarged the figure and increased the font size so that it is now readable.

Supplementary material

I appreciate the inclusion of multiple vignettes, tutorials, and external web resources linked in the paper. These additions significantly enhance the usability and accessibility of the dataset and tools, and they provide valuable guidance for potential users. Well done!
Thank you once again for the support and the valuable comments.

---

## Author Comment (AC2)

Review of "Environment90m – globally standardized environmental variables for spatial freshwater biodiversity science at high spatial resolution"

The manuscript presents Environment90m, a valuable new dataset for global freshwater research. By providing globally standardized environmental variables at high spatial resolution, this dataset addresses a key limitation in large-scale studies of freshwater ecosystems - the lack of consistent, high-resolution environmental data. The integration with the hydrographr R package is particularly commendable, as it facilitates data access and analysis at large scales.

However, before recommendation, I have several concerns regarding the clarity, structure, and presentation of the paper.

Thank you for the constructive comments which we took all into account while revising the manuscript.

Major Concerns

Manuscript structure: The structure of some sections should be revised for better logical flow (see detailed comments below).

We have done a general re-structuring of the manuscript. With this we have improved the readability and general flow of the manuscript.

Readability: Language and grammar require improvement. Sentences are frequently too long and difficult to follow. A language edit is recommended.

We checked the manuscript with special attention to grammar and improvement of the sentences to make them shorter, clearer and easy to follow.

hydrographr package updates: The extent of modifications and extensions made to the hydrographr R package is unclear. A concise summary of newly added functions - beyond those demonstrated in the case study - would be highly valuable to researchers.

We added in the "Accessing the Environment90m dataset" section (2.4) a table with a description of the new functions now available in the hydrographr R-package.

Minor Concerns

Citations often appear incorrectly formatted or are replaced by "?", suggesting broken links to the bibliography. These should be checked and corrected.

All references have been checked and corrected. The issue was due to Laxex-libraries which did not insert (the correct) citations in the pdf document.

The vignette link is not functional. I suggest including "Environment90m" in the vignette title (e.g., "Case study - Danube Basin (Environment90m)") to make it more easily discoverable.

Now the link to the vignette is working fine. We have renamed the title to the suggested one by the reviewer "Case study - Danube Basin (Environment90m)". We agree that it increases the chance to be discovered and used.

In the case study (vignette), the paths (working directory setup) appear inconsistent and worked only after adjustments; also, "flow" may need to be replaced by "accumulation" in the function:

Thank you for spotting this, there was a trailing slash "/" missing when setting up the working directory path. This has been fixed now.

download_hydrography90m_tables(subset = c("flow --> accumulation?", "length", "slope_grad_dw_cel"),
…)

The abbreviation of "flow" has been changed to "accumulation".

Section-Specific Comments

Introduction

Line 23 - 30: This paragraph is hard to read and it seems establishing a baseline is a major motivation to assemble this data set. I suggest to elaborate on this, as it is not quite clear to me what this baseline is referring to.
We split the sentences and checked the legibility again. We also added more text on the baseline, which now reads as follows: "Addressing this question requires at minimum a detailed baseline of the present-day spatial distribution of the environmental characteristics of freshwater habitats. Only after establishing a baseline, the environmental changes, or changes in biodiversity can be measured and quantified."
We hope that this resolves the question, and we are happy to add more text if needed.

Paragraph 2 (Lines 31 - 54): This section currently focuses on methodological difficulties rather than the broader relevance of the dataset. Consider moving this discussion to the Calculation section. Instead, emphasize the scientific and practical value of integrating stream networks with climatological, land cover, and soil data. I.e., I suggest to focus less on the technical challenges and more on the possibilities once these challenges are overcome.

Thank you for this comment. This section reflects our experiences in having to convince others why the choice of spatial units and the spatial resolution matters in spatial freshwater biodiversity science (it may seem more than obvious to others). We elaborate on the network continuum and explain how the environmental data has to be attributed to the respective stream

segments and sub-catchments, but it is true that we do not explain the rationale of this for each variable category. If this is needed, we are happy to add more text. Currently we explain the advantages in the Discussion and if needed.

Paragraph 3 (Lines 55 - 60): The comparison with existing datasets is useful but remains vague. Clarify how Environment90m advances beyond these products and articulate the specific knowledge gains it enables.

We added a paragraph to highlight the advantages of Environment90m over the existing datasets. Environment90m takes advantage of the added values provided by the Hydrography90m dataset. In particular, the detailed delimitation of upstream sub-catchments and therefore the availability of environmental data for these areas, normally absent in global and regional studies.

Line 81: The link is not working.

The link has been updated and is now functional (https://glowabio.github.io/hydrographr/articles/case_study_lakes.html).

Environmental Data

Section 2.1: The stream network data are already described in the Hydrography90m publication and may not need to be reiterated here. A concise reference to that paper may suffice.

We suggest to leave the description and table of the Hydrography90m dataset as it is right now, mainly to have consistency between this dataset and the other datasets. Also, it would facilitate users of the dataset or the R package to have one unique reference to find the description, and standard abbreviations of the underlying variables and datasets used. Finally, this paper can be considered stand-alone by including a brief description of the Hydrography90m data, which forms the backbone of the Environment90m dataset.

Section 2.2: Elaborate on the use of these 3 GCMs and why a combination of three models was Used.

We made a selection of 3 global circulation models (GCMs), where we selected 3 SSPs and two time periods. The rationale behind this selection is that the pair-wise combination of GCMs-SSPs for the two time periods covers a wide spectrum of short and long term future projections, considering different impacts given social, technological, economical and environmental changes. We believe that this diversity of options allows capturing and evaluating the uncertainties in studies that employ climate projections. We included this explanation when introducing the CHELSA dataset. In the conclusions we also mention future developments of the database where we expect to use more GCMs and also include the 2011-2040 period as well.

Section 2.3: How were the 22 categories selected from the original 37 ESA categories? Does the land-use data have a temporal resolution (for the years 1992 to 2020)? Please be consistent when referring to land-use or land-cover data.

The 22 categories were selected based on their consistency in global coverage over the entire time period outlined in the Land Cover CCI product user guide (CCI, 2017).
CCI, E. L. C.: Product user guide version 2.0, UCL-Geomatics: London, UK, 685, 2017.

The original categories of this dataset have been created in a hierarchical manner with different levels. Level 2 is meant to be used at regional scales and level 1 at the global scale. We adopted level 1 categories but added the following explanation in the caption of the table to clarify how the categories level 2 were aggregated: "...the numbers within parenthesis in the description make reference to the coding of the categories at level 2".

In the caption of the land cover table we explain that the numbers within the parenthesis in the description of each category are the codes taken from the original land cover categories used to aggregate the final categories. Also, in the description of the dataset, the temporal resolution is specified, as annual resolution, and therefore the data can be retrieved for each year within the 1992-2020 period. We also fixed the term to land cover data throughout the manuscript.

Section 2.4: Why did you decide to integrate over all available soil depths?

The original data is provided in different soil depths. At first we needed to decide if we prepare the data per soil depth or calculate the average. The calculation per soil depth would have been very computationally intense and selecting only one depth would generate some potential bias for each of the soil variables. Therefore, we decided to calculate the weighted average for each soil property following the GlobalSoilMap specifications as described in Arrouays et al. (2014).

Section 2.6: I suggest to also provide the variance of stream flow over the selected time period.

We agree that providing the variance of the selected time period and also the yearly estimates would be very beneficial for freshwater research. Given that the time-frame for calculations is long, we will leave these variables out for this version of the Environment90m dataset, but we will include them, among other new variables (see at the end of the discussion section), once the new calculations are finished.

Section 2.7: How exactly was AI and PET modeled? Please elaborate on the process.

We are aware that the underlying datasets and how they were created is important for potential users. At the same time, the descriptions of how the CHELSA climate, soilgrids, FLO1K and AI/PET were created would be beyond the scope of our manuscript. We opted for short descriptions in the table which briefly describes how the variables were calculated.  We processed AI and PET using the resampling procedure explained in section 2.1 to calculate the summary statistics for these variables.

Calculations

Include a short discussion on how the varying spatial resolutions of the underlying datasets affect Environment90m applications and interpretation - particularly in small headwater catchments.

This discussion is very important and we have added text in the Discussion section to develop these ideas. We discussed the advantages of aggregating the underlying datasets (i.e. calculating the summary statistics) in comparison to using the original values, in terms of variable standardization and saving computing times for regional and global analysis. This directly also took us to discuss the advantages of using the sub-catchments instead of cell grids as units of analysis given the ecological significance of the sub-catchments for freshwater biodiversity. We finally highlighted the accessibility of high resolution environmental data available in Environment90m for headwater sub-catchments, which are normally neglected or aggregated to larger basins, and open a window of research to investigate the relevance of these areas for freshwater conservation.

Use a consistent notation for spatial resolution (either 90 m or 90 m²).

We have changed all resolution notation to e.g. 90m.

Case Study Workflow

The vignette link is not working.
We have corrected the link to the vignette
https://glowabio.github.io/hydrographr/articles/case_study_lakes.html.

The case study is an excellent addition. However, please include a (short) dedicated section summarizing the new functions added to hydrographr for handling Environment90m data, possibly including a summary table.

We decided, based also on the suggestions from Reviewer 1, to include the description of the new functions of the hydrographr R-package, in the section: "Accessing the Environment90m dataset". We also included a summary table of the functions (Table 8) following the same design as in the hydrographr package manuscript.

Conclusion

It is not always clear whether the studies cited used Hydrography90m or Environment90m. Please clarify.

We have now a dedicated section (i.e., section: "Applications") to showcase the studies where the Environment90m dataset has already been used. We clarified the narratives by explaining the data used as inputs for this case studies are coming from the Environment90m datasets and that in some cases they retrieved from the large tables some of the variables originally available in the Hydrography90m dataset, as referred in Table 1. We wrote this more clearly in the text.

New functions for lake processing are introduced only here; these should be documented earlier in a dedicated section.

We added to the manuscript a new dedicated section to illustrate a second case study related to lake processing (i.e., section: Case study: environmental characterization of lakes' upstream basins). In this workflow we describe the new functions available in the hydrographr Rpackage for lake processing and show how to use Environment90m to environmentally characterize the upstream basins of the stream network and lake intersections and the upstream basin of the lake outlet.

Figures and Tables

Tables 1–7: Ensure uniform font size.
We have changed the fonts of all tables to have the same size.

Figure 1 & 3: Captions should be more descriptive and self-explanatory.
The caption of these two figures have been improved by expanding the descriptions of the workflows.

Overall Assessment

Environment90m represents an important contribution to global freshwater biodiversity science. With clearer presentation, improved language, and clearer documentation of the newly added hydrographr functions, this dataset will likely become a foundational resource for future large-scale aquatic research.
Thank you once again for the very positive and constructive comments, which helped us to improve the manuscript.

Citation: https://doi.org/10.5194/essd-2025-399-RC2